# DIFFTACTILE: A PHYSICS-BASED DIFFERENTIABLE TACTILE SIMULATOR FOR CONTACT-RICH ROBOTIC MANIPULATION

**Zilin Si** [*, †, 1,5], **Gu Zhang**[*,2], **Qingwei Ben**[*,3], **Branden Romero**[4], **Zhou Xian**[1],
**Chao Liu**[4], **Chuang Gan**[5,6]
[1] CMU RI, [2] Shanghai Jiao Tong University, [3] Tsinghua University, [4] MIT CSAIL,
[5] MIT-IBM Watson AI Lab, [6] UMass Amherst
`zsi@andrew.cmu.edu, guzhang.sjtu@gmail.com,`
`bqw20@mails.tsinghua.edu.cn, brromero@mit.edu,`
`xianz1@andrew.cmu.edu, chaoliu@csail.mit.edu,`
`ganchuang@csail.mit.edu`

## ABSTRACT

We introduce DIFFTACTILE, a physics-based differentiable tactile simulation system designed to enhance robotic manipulation with dense and physically accurate tactile feedback. In contrast to prior tactile simulators which primarily focus on manipulating rigid bodies and often rely on simplified approximations to model stress and deformations of materials in contact, DIFFTACTILE emphasizes physics-based contact modeling with high fidelity, supporting simulations of diverse contact modes and interactions with objects possessing a wide range of material properties. Our system incorporates several key components, including a Finite Element Method (FEM)-based soft body model for simulating the sensing elastomer, a multi-material simulator for modeling diverse object types (such as elastic, elastoplastic, cables) under manipulation, a penalty-based contact model for handling contact dynamics. The differentiable nature of our system facilitates gradient-based optimization for both 1) refining physical properties in simulation using real-world data, hence narrowing the sim-to-real gap and 2) efficient learning of tactile-assisted grasping and contact-rich manipulation skills. Additionally, we introduce a method to infer the optical response of our tactile sensor to contact using an efficient pixel-based neural module. We anticipate that DIFFTACTILE will serve as a useful platform for studying contact-rich manipulations, leveraging the benefits of dense tactile feedback and differentiable physics. Code and supplementary materials are available at the project website[1].

## 1 INTRODUCTION

In the goal of enabling robots to perform human-level manipulation on a diverse set of tasks, touch is one of the most prominent components. Tactile sensing, as a modality, is unique in the sense that it provides accurate, fine-detailed information about environmental interactions in the form of contact geometries and forces. Although its efficacy has been highlighted by prior research, providing crucial feedback in grasping fragile objects (Ishikawa et al., 2022), enabling robots to perform in occluded environment (Yu & Rodriguez, 2018), and detecting incipient slip (Chen et al., 2018) for highly reactive grasping, there are still advances in tactile sensing to be made especially in the form of simulation.

Physics-based simulation has become a significant practical tool in the domain of robotics, by mitigating the challenges of real-world design and verification of learning algorithms. However, existing robotic simulators either lack simulation for tactile sensing or limit interactions to rigid

---

[*] Authors with equal contribution.

[†] This work was done during an internship at the MIT-IBM Watson AI Lab.

[1] `https://difftactile.github.io/`

bodies. To accurately simulate tactile sensors which are inherently soft, it is essential to model soft body interaction's contact geometries, forces, and dynamics. Prior work (Si & Yuan, 2022) attempted to simulate contact geometries and forces for tactile sensors under (quasi-)static scenarios, and it was successfully applied to robotic perception tasks such as object shape estimation (Suresh et al., 2022), and grasp stability prediction (Si et al., 2022). However, highly dynamic manipulation tasks have not been thoroughly explored. Other prior works approach contact dynamics by either approximating sensor surface deformation using rigid-body dynamics (Xu et al., 2023) or using physics-based soft-body simulation methods such as Finite Element Method (FEM) (Narang et al., 2021). However, these methods are still limited to manipulating rigid objects.

In this work, we aim to build a differentiable tactile simulator, DIFFTACTILE , that supports contact-rich robotic manipulation of rigid, deformable, and articulated objects. Differentiability, as a key component of our work, provides fine-grained guidance for efficient skill learning (Huang et al., 2021; Xian et al., 2022). It also enables system identification to close the sim-to-real gap (Li et al., 2023). We implement DIFFTACTILE in Taichi (Hu et al., 2019) which leverages parallel GPU computing and auto-differentiation. To demonstrate the capability and versatility of

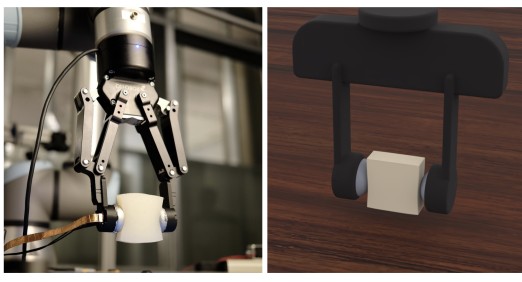

**Figure 1:** Grasping a deformable object in the real world and in DIFFTACTILE.

our simulator, we evaluate it on a diverse set of manipulation tasks including handling fragile, deformable, dynamic objects that cannot be addressed with prior tactile simulators. We summarize our contributions below:

- We introduce DIFFTACTILE , a platform supporting various tactile-assisted manipulation tasks. We model tactile sensors with FEM, objects in various materials (rigid, elastic, and elastoplastic) with Moving Least Square Material Point Method (MLS-MPM), and cable with Position-Based Dynamics (PBD). We simulate the contact between sensors and objects with a penalty-based contact model. In addition, we accurately simulate the optical response of tactile sensors with high spatial variation via a learning-based method.

- Our system is differentiable and can reduce the sim-to-real gap with system identification. From a sequence of real-world data samples, we can optimize our simulator's sensor material and contact model parameters with differential physics and validate it with more general real-world scenarios.

- We demonstrate the improvement of skill learning efficiency with tactile feedback. We evaluate it on stable and adaptive grasps of objects with diverse geometry and material properties, and four contact-rich manipulation tasks.

## 2 RELATED WORK

**Tactile simulation** The most recent work on tactile simulation is built upon existing rigid-body simulators. For example, Tacto (Wang et al., 2022), Tactile-Gym (Church et al., 2022; Lin et al., 2022) were built upon PyBullet (Coumans & Bai, 2016). An efficient tactile simulation (Xu et al., 2023) was built upon DiffRedMax (Xu et al., 2021), where a penalty-based contact model was used to simulate the force distribution for tactile sensors. Even though it is computationally efficient to use rigid body simulation, these tactile simulators approximate contact dynamics for soft bodies at the cost of fidelity.

Alternatively, Finite Element Method (FEM)-based methods exist to accurately simulate soft body dynamics. A physics-based tactile simulator (Narang et al., 2021) was developed for SynTouch BioTac sensors (SynTouch) by using FEM in Isaac Gym (Makoviychuk et al., 2021). A grasp simulator also used the FEM in Isaac Gym (Kim et al., 2022) with incremental potential contact (IPC) model to handle contact dynamics. Taxim (Si & Yuan, 2022) used a superposition method to approximate the FEM. We also model tactile sensors with FEM to maintain the simulator's physical accuracy and extend the contact model to handle objects with various materials beyond rigid.

**Differentiable physics-based simulation** Differentiable physics-based simulation has become popular in recent years as it allows for efficient gradient-based policy learning compared to

| Tactile simulator | Object model | | Backend | Method | Optical | Differentiability |
| --- | --- | --- | --- | --- | --- | --- |
| | Rigid | Soft | | | Simulation | |
| Tacto (Wang et al., 2022) | ✓ | | PyBullet | Rigid body | ✓ | |
| (Xu et al., 2023) | ✓ | | DiffRedMax | Rigid body | ✓ | ✓ |
| Tacchi (Chen et al., 2023) | ✓ | | Taichi | MPM | ✓ | |
| Taxim (Si & Yuan, 2022) | ✓ | | PyBullet | FEM | ✓ | |
| (Narang et al., 2021) | ✓ | | Isaac Gym | FEM | | |
| IPC-GraspSim (Kim et al., 2022) | ✓ | ✓ | Isaac Gym | FEM | | |
| **Ours** | ✓ | ✓ | Taichi | FEM | ✓ | ✓ |

**Table 1:** Comparison with other state-of-the-art tactile simulators. We show that DIFFTACTILE is the only tactile simulator supporting simulating objects with various materials while being system-wise differentiable and physically accurate.

traditional sampling-based algorithms. PlasticineLab (Huang et al., 2021), FluidLab (Xian et al., 2022), SoftZoo (Wang et al., 2023) were presented with differentiability for soft body manipulation, fluid manipulation, and soft robot co-design, respectively, by leveraging Moving Least Square Material Point Method (MLS-MPM) (Hu et al., 2018). Tacchi (Chen et al., 2023) also used MLS-MPM to simulate the soft body deformation for GelSight (Yuan et al., 2017), a type of vision-based tactile sensor but did not present differentiability and contact dynamics modeling. It is shown that differential physics can be applied for system identification (Ma et al., 2023) to fine-tune the simulator's physical parameters and reduce the sim-to-real gaps. However, it remains unclear whether the gradient-based approach can benefit to improve the efficiency of tactile-assisted manipulation skill learning.

**Optical Simulation**   Taxim (Si & Yuan, 2022) showed that data-driven approaches to simulate the optical response of vision-based tactile sensors significantly outperform model-based methods such as (Wang et al., 2022; Chen et al., 2023; Agarwal et al., 2021; Gomes et al., 2021). However, there is a divergence in data-driven approaches. Previous work including (Higuera et al., 2023; Chen et al., 2022; Zhong et al., 2023) use image generation techniques like generative models to perform style transfer from a simulated image to the style of a real deformation. However, these methods are rather data-intensive since they need a large variation of real-world examples to generalize well. Instead, Taxim (Si & Yuan, 2022) takes a pixel-based approach that uses a polynomial lookup table to map surface normals to RGB directly. It is more data-efficient but makes assumptions about the sensors bidirectional reflectance distribution function (BRDF), which limits its applicability to sensors with low spatial variance.

We compare our work with state-of-the-art tactile simulators in Table 1. We show that our work, to the best of our knowledge, is the only work that is 1) system-wise differentiable to enable efficient skill learning, 2) can accurately model the soft body dynamics and contact dynamics, 3) supports broad categories of objects including rigid, elastic, elastoplastic, and cables, and 4) provide a data-efficient approach to simulate optical responses for vision-based tactile sensors.

## 3 TACTILE SIMULATION

### 3.1 SYSTEM OVERVIEW

DIFFTACTILE models the soft contact between tactile sensors and objects including contact force distribution, contact surface deformation, and optical response to provide dense tactile feedback. We present four key modules of our system: 1) a Finite Element Method (FEM)-based tactile sensor model in Section 3.2, 2) a learning-based method to simulate the optical response of tactile sensors with high spatial variation in Section 3.3, 3) rigid, elastic, and elastoplastic object models using Moving Least Square Material Point Method (MLS-MPM), and cable model using Position-Based Dynamics (PBD) in Section 3.4, 4) a penalty-based contact model in Section 3.5.

## 3.2 TACTILE SENSOR SIMULATION

We model the deformation of the tactile sensor's soft elastomer under contact forces with FEM. We discretize the sensor soft elastomer to tetrahedron elements and then apply boundary conditions at the base of the sensor with position or velocity control. Since most tactile sensors' elastomers including ours are made from hyper-elastic materials, we apply the Neo-Hookean constitutive model in our simulation to capture the non-linearity of the material property. The energy density function $\Psi$ and the first Piola-Kirchhoff stress tensor $\mathbf{P}$ used for governing equations are defined as:

$$\Psi(I_1, J) = \frac{\mu}{2}(I_1 - 3) - \mu log(J) + \frac{\lambda}{2}log^2(J)$$
$$\mathbf{P}(\mathbf{F}) = \mu(\mathbf{F} - \mathbf{F}^{-\mathbf{T}}) + \lambda log(J)\mathbf{F}^{-T}$$

(1)

where $\mathbf{F} \in \mathbf{R}^{3\times3}$ is the deformation gradient, $I_1 = tr(\mathbf{F}^T\mathbf{F})$ is the first isotropic invariants, and $J = det(\mathbf{F})$ is an additional invariant. Note that our tactile simulation can be easily customized with different shapes, sizes, and materials by replacing the input mesh model or constitutive model.

To get tactile outputs including visual images and marker motions for vision-based tactile sensors, we first extract the deformed surface mesh from each simulation step's FEM solution, then we interpolate the marker's locations by weighting surface node locations given a set of initial markers captured from a real sensor. We project 3D markers to the 2D image plane by using the tactile sensor's camera model.

## 3.3 OPTICAL SIMULATION

We reconstruct the optical response of a vision-based tactile sensor to contact using a data-driven approach. We model the surface of the sensor as a height function $z = f(x, y)$, and represent the continuous spatially-varying reflectance function of the surface as a 4D vector-valued function. The function input is the 2D viewing direction (d $= \theta, \varphi$) and 2D surface normals (x $= \frac{\partial f}{\partial x}, \frac{\partial f}{\partial y}$), and the output is the change in reflected color $c = (r, g, b)$. We approximate our reflectance function with a multilayer perceptron (MLP) $f_\theta$ whose input is augmented with a positional encoding $\gamma(d)$ and $\gamma(x)$ rather than directly $d$ and $x$ to enable the network to better fit data with high-frequency variation (Mildenhall et al., 2021). Formally the encoding function is:

$$\gamma(p) = \sin(2^0\pi p), \ \cos(2^0\pi p), \ ..., \ \sin(2^{L-1}\pi p), \ \cos(2^{L-1}\pi p)$$

(2)

Our rendering scheme finally consists of approximating the deformation caused by the contact indentation using pyramid Gaussian kernels as proposed in (Si & Yuan, 2022).

## 3.4 OBJECT SIMULATION

We aim to support broader categories of objects beyond rigid objects for more diverse manipulation applications. We leverage the Moving Least Square Material Point Method (MLS-MPM) (Hu et al., 2018) to simulate rigid, elastic, elastoplastic objects. MLS-MPM has been shown to be efficient in simulating soft bodies. For elastic objects, we implement both corotated linear elasticity and Neo-Hookean elasticity models. For elastoplastic objects, we use the von Mises yield criterion to model plasticity upon elasticity. For rigid objects, we first treat objects as elastic using MLS-MPM, and then we add rigidity constraints by calculating object transformation and enforcing the shape of the object. For articulated objects, we approximate the simulation by using the MPM-based approach and assign different materials for different parts. The joints are simulated as soft and thin bodies and other parts are simulated as rigid bodies.

For another group of deformable objects such as cables and clothes, it is common to simulate them with Position Based Dynamics (PBD) (Müller et al., 2007). We also incorporate cable objects in our simulation by using PBD, where we constrain the stretch, bending, and self-collision.

## 3.5 PENALTY-BASED CONTACT MODEL

We handle contact dynamics between sensors and objects with a penalty-based contact model similar to (Xu et al., 2023). At each simulation step, we first check contact collision by pairing the surface

triangle mesh from FEM with surface nodes from the object's particles (with either MPM or PBD). For each pair, we calculate the sign distance field $d$ and normal directions $\mathbf{n}$ from the node to the triangle mesh. If $d$ is negative, the node is penetrating the surface mesh and we need to apply normal penalty force to both mesh nodes and particle node to constrain the contact. In addition, we apply static or dynamic friction forces to the pair based on their relative velocities and normal forces. We represent our contact model as:

$$
\begin{aligned}
\mathbf{f_n} &= -(k_n + k_d \mathbf{v_n})d\mathbf{n} \\
\mathbf{f_t} &= -\frac{\mathbf{v_t}}{||\mathbf{v_t}||}\min(k_t||\mathbf{v_t}||, \mu||\mathbf{f_n}||)
\end{aligned}
\tag{3}
$$

where $\mathbf{f_n}$ and $\mathbf{f_t}$ are contact forces in the normal and tangential direction with respect to the local surface triangle. $\mathbf{v_n}$ and $\mathbf{v_t}$ are the relative velocities between the pair of the triangle and node in normal and tangential directions. $k_n$, $k_d$, $k_t$ and $\mu$ are the parameters of contact stiffness, contact damping, friction stiffness, and friction coefficient. Then the contact force $\mathbf{f} = \mathbf{f_n} + \mathbf{f_t}$ is applied to both the triangle mesh nodes and the particle node of the pair as an external force.

**FEM-MPM coupling** FEM is a mesh-based method and we can extract surface triangle meshes along with their associated node positions, velocities and face normal directions. MLS-MPM is a meshless hybrid Lagrangian-Eulerian method that uses Lagrangian particles and Eulerian grids to simulate continuous materials. We apply contact collision checking and contact force modeling between FEM surface mesh nodes and MPM Eulerian grids for efficiency.

In each simulation step, we first pre-compute the internal elastic forces for all tetrahedral meshes from the constitutive law for the FEM sensor model, and advance particles to grids for the MPM object model. Then we check contact collision and calculate external contact forces for all pairs of triangle meshes and grid, and add them to the surface nodes. In post-contact computing, we transfer the velocities and affine coefficients from the grid to particles and do particle advection for the MPM object model; and we advect the positions and velocities of the nodes based on the internal elastic forces, external contact forces, and gravity for FEM elements. We also consider the external boundaries such as tables and walls to constrain the positions of the objects.

**FEM-PBD coupling** Similarly to FEM-MPM coupling, we simply replace the MPM particles with PBD particles for contact collision detection and modeling. For PBD objects, there's no pre-contact computation, but we need to solve the stretch, bending, and self-collision constrains after the contact, and velocity advection based on the updated positions.

## 4 EXPERIMENTS

### 4.1 OVERVIEW

We present two sets of tasks with DIFFTACTILE: system identification, and tactile-assisted manipulation. For system identification, we use real-world tactile observations to optimize the simulator's system parameters and to reduce sim-to-real gaps. Then we present five manipulation tasks: grasping, surface following, cable straightening, case opening, and object reposing as shown in Fig. 2. Tactile sensing can enable safer and more adaptive grasping to handle fragile objects such as fruits. We grasp a diverse set of objects with various shapes, sizes, and materials without slipping and damaging. For the other four contact-rich manipulation tasks, surface following requires the sensor to stay in contact with a 3D surface and travel to an endpoint while maintaining a certain contact force; cable straightening requires a pair of sensors to first grasp a fixed end of the cable, and then straighten it by sliding towards the other end; case opening uses a single sensor to open an articulated object via pushing; lastly, object reposing involves using a single sensor to push an object from a lying pose to a standing pose against the wall. These four tasks represent rigid, deformable, and articulated object manipulation.

### 4.2 SIMULATION SETUP

**Initialization** We initialize the simulation environment with a single tactile sensor $s$ for system identification, surface following, case opening, and object reposing, and two tactile sensors $\{s_1, s_2\}$ mounted on a parallel jaw gripper for grasping and cable straightening. Both tactile sensors' and

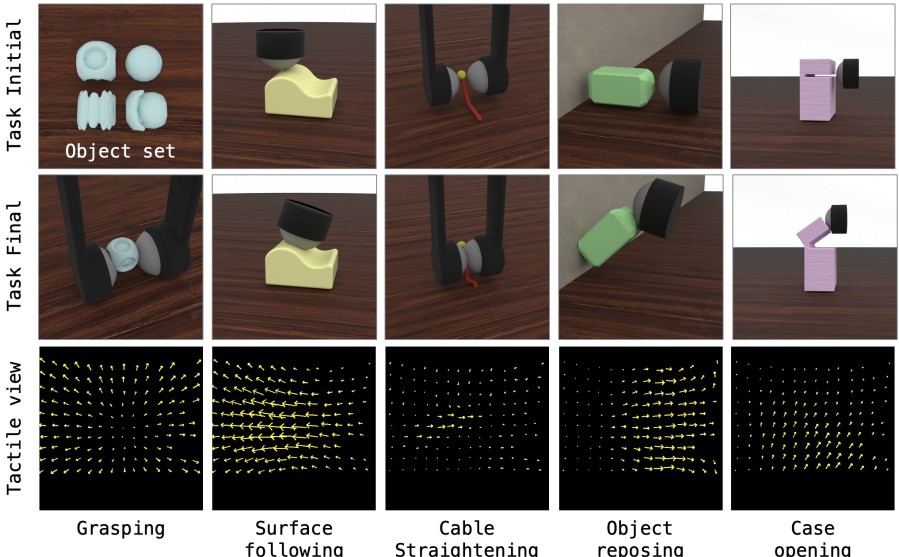

**Figure 2:** DIFFTACTILE tasks. **Grasping**: We grasp a set of four objects with different geometries and materials. **Surface following**: A sensor travels on the surface while maintaining the contact. **Cable straightening**: A pair of sensors straighten a cable by gripping and sliding from a fixed end. **Object reposing**: A sensor pushes an object to let it stand against a wall. **Case opening**: A sensor opens the cap of a case.

objects' shapes are initialized with STL or OBJ mesh models and then voxelized to FEM tetrahedron meshes or MPM/PBD particles. Objects $o_i$ are initialized statically on the tabletop and we add a vertical wall for object reposing. Tactile sensors are initialized statically near objects depending on tasks but without contact. We initialize the poses of tactile sensor at time step $t = 0$ as $T_s(0) = (R_s(0), t_s(0)) \in SE(3)$ where $R_s(0) \in SO(3)$ and $t_s(0) \in \mathbb{R}^3$ and similarly object pose as $T_o(0)$.

**State** Each tactile sensor $s$ is represented as an FEM entity with $N$ nodes and $M$ tetrahedral elements. For each node $n_i$, it contacts a 6D state vector $s_i(t) = \{p_i(t), v_i(t)\}$ including a 3D position $p_i(t)$ and a 3D velocity $v_i(t)$. For each element $m_i$, it contacts a 4D index mapping from the element to its associated four nodes. Both MPM-based and PBD-based objects are represented with particles, and similarly, each particle $o_i$ also has a 6D state vector $o_i(t) = \{p_i(t), v_i(t)\}$.

**Observation** We define two types of observations of each simulation step $t$, the state observation and the tactile observation. State observation includes tactile sensors' and objects' poses $T_s(t), T_o(t)$ and each node's or particle's state $s_i(t), o_i(t)$. For tactile observation, we can output the sensor's surface triangle mesh as a deformation map, the sensor's surface force distribution, or an aggregated three-axis force vector.

**Action** At each time step $t$, actions for end-effectors (either tactile sensors or gripper with kinematic chains down to tactile sensors) are queried from the controller as represented as a velocity vector $v_s(t) = \{\Delta R_s(t), \Delta t_s(t)\}$ to update the velocities of the FEM nodes.

**Reward/Loss** Each task's reward or loss function is formed differently based on the task objectives. We refer the readers to Section A.4 for more details.

## 4.3 SYSTEM IDENTIFICATION

Sim-to-real transfer for robot learning has been a long-standing challenge where the gap in between heavily relies on simulation fidelity. To reduce the gap, we leverage differentiable physics to optimize the physical parameters of material and contact models given example data from the real world. Our optimization targets include Lamé parameters $\mu$ and $\lambda$ of the FEM sensor model, and $k_n, k_d, k_t, \mu$ of the contact model. The optimization objectives include the 6-axis force readings and tactile marker readings under four different contact scenarios: pressing, sliding, in-plane twisting, and tilt twisting as shown in Fig. 3.

**Experimental Setup and Dataset** We collect sequences of contact data from both the real world and simulation with synchronized control poses and velocities of the sensor.

|  |  | Press-slide↓ | Press-twist-z↓ | Press-twist-x↓ |
|---|---|---|---|---|
| **Sim2Sim** | **Random** | $1.69 \pm 1.10$ | $1.15 \pm 0.51$ | $1.43 \pm 0.62$ |
|  | **RNN** | $1.20 \pm 0.42$ | $0.68 \pm 0.28$ | $0.90 \pm 0.26$ |
|  | **CMA-ES** | $0.59 \pm 0.12$ | $0.47 \pm 0.15$ | $0.61 \pm 0.20$ |
|  | **Ours** | $\mathbf{0.53 \pm 0.35}$ | $\mathbf{0.42 \pm 0.24}$ | $\mathbf{0.58 \pm 0.28}$ |
| **Real2Sim** | **Random** | $3.54 \pm 1.73$ | $2.59 \pm 0.99$ | $4.47 \pm 3.31$ |
|  | **RNN** | $3.29 \pm 1.51$ | $2.42 \pm 0.90$ | $4.53 \pm 3.51$ |
|  | **CMA-ES** | $3.42 \pm 1.47$ | $2.67 \pm 1.22$ | $4.99 \pm 4.30$ |
|  | **Ours** | $\mathbf{3.08 \pm 1.27}$ | $\mathbf{2.38 \pm 0.86}$ | $\mathbf{3.99 \pm 2.89}$ |

**Table 2:** The pixel-wise tactile marker mean squared errors with standard deviation to evaluate system identification.

As shown in Fig. 3, there are three types of data sequences, *press-slide*, *press-twist-z* (twist along the z-axis), and *press-twist-x* (twist along the x-axis). For this experiment, the sensor interacts with two surfaces with different frictional properties, acrylic and tape.

**Experimental results** We evaluate two sets of experiments: **Sim2Sim** and **Real2Sim** where we use simulated data or real data respectively as inputs of the system. We optimize the sensor and contact model parameters with *press-slide* sequence and test on all three sequences. We compare gradient-based trajectory optimization (**Ours**) with three baselines, **Random**, **RNN**, and **CMA-ES** as shown in Table 2. Here we use pixel-wise mean squared error (MSE) between predicted and collected tactile

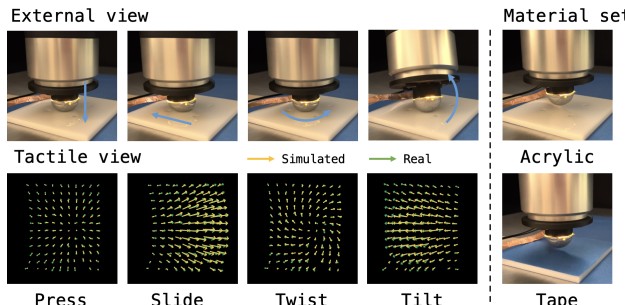

**Figure 3:** System identification to optimize the FEM sensor model and contact model's physical parameters with tactile readings and force readings from the real world.

markers as evaluation metrics. For **Random**, we randomly select parameters within a practical range; for **RNN**, we input tactile marker readings and force readings and output the predicted system parameters; for **CMA-ES**, we sample predicted parameters from algorithm's distribution function. **Ours** outperforms **Random**, **RNN** and **CMA-ES** on all sequences for both **Sim2Sim** and **Real2Sim**.

We use the identified tactile sensor parameters from **Real2Sim** for all following manipulation tasks. However, contact parameters such as the surface friction coefficient also depend on object materials. But these can still serve as good references and we use them by adding randomization based on the identified parameters. For object parameters, we randomize them within a range based on the tactile sensor's and contact model's parameters to make sure the system can stably run.

## 4.4 OPTICAL SIMULATION

**Experimental setup and dataset** We manually collect 250 example deformations across the entire sensing surface using a 4mm spherical indenter from the real world. The pose of the sphere is manually annotated, and we split the dataset into a training set consisting of 200 examples, with the rest held out for testing.

**Experimental results** We test our method against a polynomial table mapping from Taxim (Si & Yuan,

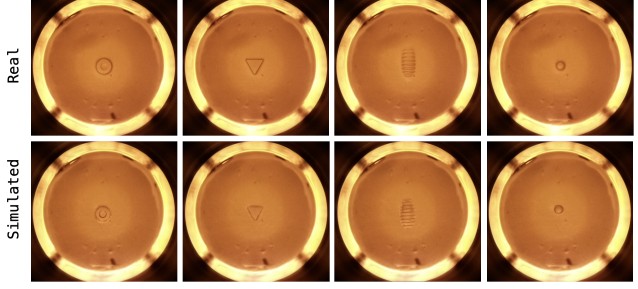

**Figure 4:** Tactile optical simulation compared with real data capturing various contact geometries.

|       | L1↓  | MSE↓  | SSIM↑ | PSNR↑ |
|-------|------|-------|-------|-------|
| **Taxim** | 16.1 | 85.74 | 0.998 | 38.47 |
| **Ours**  | **7.94** | **56.10** | **0.999** | **39.42** |

**Table 3:** Image similarity metrics for our test set. We compare our method to Taxim (Si & Yuan, 2022) on L1, MSE, SSIM and PSNR metrics. Our method performs better across all metrics.

|       | Loss | $L_{pos}$↓ | $D_{slip}$↓ | $L_{deform}$↓ |
|-------|------|-----------|-------------|---------------|
| **Elastic** | **w/o tactile** | $0.04 \pm 0.03$ | $0.18 \pm 0.06$ | N/A |
|       | **w/ tactile** | $\mathbf{0.01 \pm 0.01}$ | $\mathbf{0.07 \pm 0.04}$ | N/A |
| **Elasto-plastic** | **w/o tactile** | $0.70 \pm 0.53$ | $0.26 \pm 0.09$ | $\mathbf{0.85 \pm 0.25}$ |
|       | **w/ tactile** | $\mathbf{0.24 \pm 0.01}$ | $\mathbf{0.2 \pm 0.05}$ | $0.89 \pm 0.48$ |

**Table 4:** Evaluation of grasping deformable, fragile objects with position, deformation losses, and slipping distance by either using or not using tactile observations.

2022). We use pixel-wise MSE, L1, SSIM, and PSNR as evaluation metrics. As shown in Table 3, our method outperforms Taxim across all metrics. Additionally, we verify the generalization and accuracy of our method by rendering a set of test probes with varying geometry, along with example real-world indentations for comparison in Fig. 4. We show our method can capture contact geometries in great detail.

## 4.5 GRASPING

**Experimental setup and dataset**   We evaluate our simulator on grasping objects with various object properties including different shapes, sizes, weights, and material properties. As shown in Fig. 2, we select four objects from EGAD (Morrison et al., 2020) dataset with different shape complexity and assign each object with two different material properties, elastic, and elastoplastic.

We aim to grasp objects stably and adaptively to avoid slipping and damaging the object with gradient-based trajectory optimization. Here we use two tactile sensors as fingertips and mount them on a parallel jaw gripper. In each trajectory, the gripper first grips the object and then lifts it. Based on our goal, we define the objectives with three types of losses 1) **Position loss** $L_{pos}$: we set a 3D target position to reach after lifting; 2) **Deformation loss** $L_{deform}$: we aim to keep the shape of the object during the grasp by using the sign distance field of the object and the L1 distance of the mass distribution between the current object and the target one to penalize the deformation (Huang et al., 2021) 3) **Slipping loss** $L_{slip}$: we use the shear force detected between the fingertip and the object to penalize the slippage during grasping.

**Experimental results**   We evaluate the grasping with or without tactile feedback on three metrics. We use $L_{pos}$ for both types of objects, and we use $L_{deform}$ for elastoplastic objects only. In addition, we measure the slipping distance of the object relative to the sensor for both sets of objects, the slipping distance is denoted as $D_{slip}$. We show in Table 4 that the tactile feedback greatly improves the grasping quality.

## 4.6 CONTACT-RICH MANIPULATION

**Experimental setup**   For all four manipulation tasks, we define two different rewards, state reward and tactile reward for manipulation skill learning. We evaluate our system's learning efficiency by comparing gradient-based trajectory optimization with a sampling-based trajectory optimization, CMA-ES (Hansen et al., 2003), and model-free RL algorithms, SAC (Haarnoja et al., 2018), and PPO Schulman et al. (2017).

**Surface following**   We set up a sensor to travel and follow a curved 3D surface. We define the state reward as traveling to a certain position on the 3D surface, and the tactile reward as keeping contact with the surface while maintaining a constant shear motion.

**Cable straightening**   We set up a parallel jaw gripper with two tactile fingers and a cable with one end fixed to the wall while the other end is free. The state reward is defined as the distance between the target position (the cable is horizontally straight) and the current position for each node on the

| | Obs | Rew | Manipulation tasks | | | |
|---|---|---|---|---|---|---|
| | w/ tac | w/ tac | ObjectRepose ↑ | CableStraighthen ↓ | CaseOpen ↑ | SurfaceFollow ↑ |
| **PPO** | × | × | $4.57 \pm 0.06$ | $2.06 \pm 0.00$ | $-0.95 \pm 0.04$ | $1.33 \pm 1.53$ |
| | ✓ | × | $4.49 \pm 0.08$ | $2.07 \pm 0.02$ | $-0.93 \pm 0.26$ | $1.67 \pm 0.58$ |
| | × | ✓ | $4.64 \pm 0.15$ | $2.03 \pm 0.04$ | $-0.83 \pm 0.06$ | $2.67 \pm 1.15$ |
| | ✓ | ✓ | $4.30 \pm 0.15$ | $1.90 \pm 0.18$ | $-0.80 \pm 0.24$ | $1.33 \pm 1.53$ |
| **SAC** | × | × | $5.00 \pm 0.01$ | $1.50 \pm 0.02$ | $-0.68 \pm 0.29$ | $11.00 \pm 0.00$ |
| | ✓ | × | $4.90 \pm 0.01$ | $2.03 \pm 0.02$ | $-0.89 \pm 0.09$ | $10.00 \pm 5.57$ |
| | × | ✓ | $4.89 \pm 0.11$ | $1.60 \pm 0.12$ | $-0.84 \pm 0.04$ | $14.00 \pm 2.00$ |
| | ✓ | ✓ | $4.68 \pm 0.11$ | $1.36 \pm 0.03$ | $-0.95 \pm 0.07$ | $1.33 \pm 1.53$ |
| **CMA-ES** | N/A | × | $4.65 \pm 0.14$ | $1.97 \pm 0.14$ | $-1.07 \pm 0.05$ | $2.33 \pm 1.15$ |
| | N/A | ✓ | $4.50 \pm 0.05$ | $1.97 \pm 0.15$ | $-0.98 \pm 0.07$ | $1.67 \pm 1.15$ |
| **Ours** | N/A | × | $12.07 \pm 12.46$ | $1.27 \pm 0.81$ | $\mathbf{17.11 \pm 0.05}$ | $4.00 \pm 1.00$ |
| | N/A | ✓ | $\mathbf{60.82 \pm 0.00}$ | $\mathbf{0.89 \pm 0.32}$ | $9.83 \pm 0.38$ | $\mathbf{51.67 \pm 12.86}$ |

**Table 5:** Evaluation of manipulation tasks by comparing gradient-based optimization (**Ours**) with sampling-based optimization (**CMA-ES**), and reinforcement learning approaches (**SAC**, **PPO**).

cable. The tactile reward is defined as the force applied to the cable to maintain the gripping while being able to slide along the cable.

**Case opening** We initialize a closed case and we use a tactile sensor to push and open the lid of the case. We define the state reward as the angle of the opened lid and the tactile reward as the push forces to open the lid.

**Object reposing** A block is placed flat on the table and we aim to use one tactile sensor to flip it 90 degrees and make it stand against a wall. We define the state reward as the angle between the object and the floor, and the tactile reward as the push forces to flip the object.

**Experimental Results** To evaluate the performance of trained policies for different tasks, we design task-specific evaluation metrics: We use the traveling distance of the sensor in contact with the surface for the surface following task; the aggregation distance between the current and target cable nodes' locations for the cable straightening task; the orientation changes of the lid and the object from the beginning to the end of the trajectories for case opening and object reposing tasks.

We show all experimental results in Table 5 by comparing our proposed gradient-based optimization method with baselines. We show **Ours** outperforms baselines with a large margin to show its learning efficiency. And **w/ tactile** has better performances compared to **w/o tactile** for most tasks indicating tactile sensing helps on these contact-rich manipulation tasks.

## 5 CONCLUSIONS AND FUTURE WORK

We present DIFFTACTILE, a physics-based differentiable tactile simulator to advance skill learning for contact-rich robotic manipulation. By providing models for tactile sensors, multi-material objects, and penalty-based contacts, we greatly extend the capabilities and applicability of robotic simulators. The differentiability of our system aids in reducing the sim-to-real gaps by using system identification and improves the skill learning efficiency by providing gradient-based optimization. We evaluate DIFFTACTILE 's versatility with the grasp of a set of various objects, and manipulation tasks including surface following, cable straightening, case opening, and object reposing. By comparing with the state-of-the-art reinforcement learning and sample-based trajectory optimization approaches, we demonstrate that DIFFTACTILE can enable efficient skill learning with tactile sensing and potentially serve as a learning platform for broader tactile-assisted manipulation tasks.

In future work, we plan to integrate our tactile simulator into commonly used robotic simulation frameworks to extend its usage on more general manipulation configurations such as adding tactile sensors on dexterous robotic hands for in-hand manipulation. We would also like to investigate robot learning with multi-modalities in simulation such as leveraging vision and touch feedback to improve the robustness of the policies.

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

# A APPENDIX

## A.1 SIMULATION DETAILS

We implement our whole system with Taichi (Chen et al., 2023) along with Python to benefit from its high computing performance and auto-differentiability. With Taichi, our system can switch between running with CPU or being accelerated by GPU by simply passing an argument to initialize the Taichi environment. Taichi also supports automatic differential features for functions with explicit time integration. Therefore, considering the implementation difficulty and generalizability, our system is implemented with semi-explicit time integration, and without any extra effort, is fully differentiable and can be used for gradient-based trajectory optimization. The simulation pipeline for each simulation step can be seen in Fig. 6.

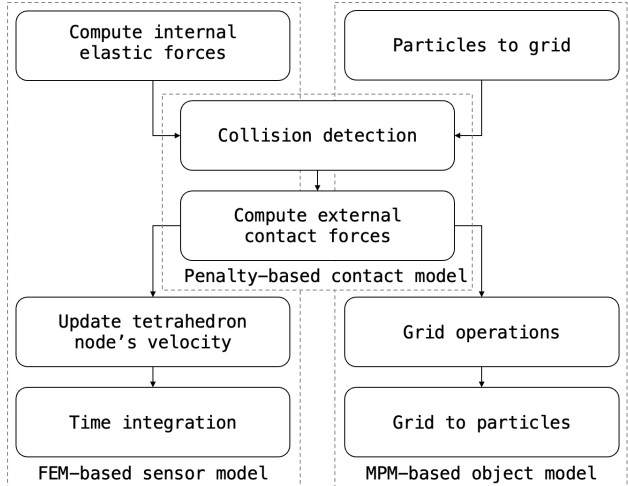

**Figure 5:** Simulation pipeline for each simulation step. Both the FEM sensor and MPM object have their pre-contact updates, and then we use a two-way coupling to handle collision and calculate contact forces. The contact forces are used in post-contact for both the FEM sensor and MPM object.

## A.2 SYSTEM IDENTIFICATION DETAILS

**Real-world data collection** We collect sequences of contact data from the real world including the 6-axis force readings from a robot arm end-effector, the poses of a Gelsight tactile sensor, and the corresponding tactile images from the tactile sensor. We set up the experiment by mounting a GelSight tactile sensor on the end-effector of an Ur5e robot arm and then controlling the robot arm to get the sensor in contact with a tabletop surface. As discussed in (Yuan et al., 2017), four general contact patterns are essential to capture and simulate for tactile sensors including contact under normal force, shear force, in-plane torque, and tilt torque. Therefore we collect three types of sequences of contact data: *press-slide*, *press-twist-z*, and *press-twist-x*. For each sequence, we start by pressing the sensor normally to a flat surface with a constant velocity of 1 mm/s for 10 seconds to get in contact. Then we slide the sensor along the surface, twist it along the normal direction, or twist it along a horizontal direction to finish *press-slide*, *press-twist-z*, and *press-twist-x* respectively with a constant velocity of 1 mm/s or 2 degrees/s for 10 seconds.

**Gradient-based estimation** We define the losses including the pixel-wised tactile marker distances as the *tactile loss* and three-axis force errors as the *force loss* between the simulated and ground truth data. Since the two losses are on different numerical scales, we aggregate them by scaling with weights 10:1 as the final loss. We use Adam optimizer with $\beta_1 = 0.9$, $\beta_2 = 0.999$. Learning rate parameters are $lr_{kn} = 20.0$, $lr_{kd} = 20.0$, $lr_{kt} = 5.0$, $lr_{fc} = 5.0$, $lr_{\mu} = 50.0$, $lr_{\lambda} = 50.0$ depending on their numerical scales. We run 100 optimization steps for each trajectory.

**RNN-based estimation** We use the Long Short-Term Memory (LSTM) model as the network architecture. The inputs of the LSTM module are with the size of $2 \times 136 + 3 = 275$, where we use 136 tracked markers' 2D motions from tactile images, and three-dimensional aggregated contact forces. We set the hidden layer size to 256 and used the default settings for other parameters. We use

a linear layer after the LSTM module with an input size of 256 and an output size of 6, to predict the six parameters of the sensor material and the contact model. We generate a simulated dataset that includes tactile marker readings, and three-axis contact force readings based on randomized system parameters. Our dataset has 2010 samples, 1800 for training, 200 for validation, and 10 for testing. Each sample's system parameters are randomized within pre-determined ranges, which ensures their real-world applicability as shown in Table 6. The model was trained in batch size of 32 for 3000 epochs, and using an Adam optimizer with a learning rate of 0.001.

| Parameter | Lower | Upper |
|---|---|---|
| $Kn$ | 10 | 100 |
| $Kd$ | 100 | 400 |
| $Kt$ | 50 | 150 |
| $Fc$ | 5 | 20 |
| $\mu$ | 800 | 1500 |
| $\lambda$ | 7000 | 10000 |

**Table 6:** Range of parameter randomization.

**Random estimation** We also provide a random estimation as our baseline. We use 10 sets of randomized system parameters from our dataset and then compare the simulated tactile markers with the ground-truth markers either from the simulation or the real world.

**Experimental details** We evaluate the system identification with simulation-to-simulation (sim2sim) and real-to-simulation (real2sim). For sim2sim, we create a dataset by randomly sampling ten sets of parameters within appropriate ranges and simulating their corresponding tactile marker data. We validate different methods including baselines and ours by predicting parameters on this dataset. The Mean Squared Error (MSE) of the marker positions between the ground truth and the simulated ones with estimated parameters is used as the evaluation metric. We compute the mean and standard deviation of these ten average marker errors for the entire trajectory.

For real2sim, we collect the dataset from the real world and apply different methods including baselines and ours to estimate the parameters. Then we simulate tactile markers using the optimized parameters and evaluate the performance by computing the mean and standard deviation of the marker errors between the actual and simulated tactile marker data.

## A.3 OPTICAL SIMULATION DETAILS

### A.3.1 NETWORK ARCHITECTURE

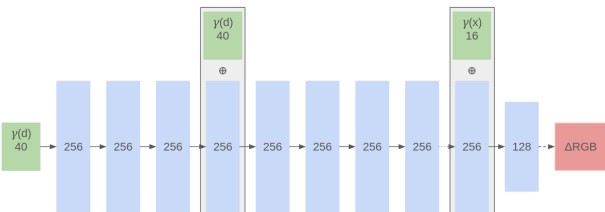

**Figure 6:** Multi layer perceptron neural network architecture used for optical simulation inspired by (Mildenhall et al., 2021). We represent inputs with green blocks, hidden layers with blue blocks, and outputs with red blocks. Solid arrows represent ReLU activation, dotted arrows means no activation, and dashed arrows mean sigmoid activation. $\oplus$ represents vector concatenation.

### A.3.2 TRAINING DETAILS

In our experiments we optimize our model using ADAM optimizer with a fixed learning rate of 1e-5 for 500 epochs. Each batch consists of all the data from a single example image. Training takes approximately 45 minutes for 200 examples.

| Lr | ObjectRepose | CableStraighten | CaseOpen | SurfaceFollow | GraspElastic | GraspPlastic |
|---|---|---|---|---|---|---|
| $lr_p$ | $5e0$ | $1e-2$ | $1e3$ | $5e-7$ | $5e-2$ | $5e-2$ |
| $lr_o$ | $1e3$ | $1e-2$ | $1e1$ | $5e-5$ | $1e-5$ | $1e-5$ |
| $lr_w$ | N/A | $1e-2$ | N/A | N/A | $5e-2$ | $5e-2$ |

**Table 7:** Learning rate of ours method in each task

| | ObjectRepose | CableStraighten | CaseOpen | SurfaceFollow | GraspElastic | GraspPlastic |
|---|---|---|---|---|---|---|
| $\alpha$ | $1e1$ | $1e-2$ | $1e1$ | $1e2$ | $1e2$ | $5e-2$ |
| $\beta$ | $5e-12$ | $1e-5$ | $5e-12$ | $1e0$ | $5e0$ | $1e1$ |

**Table 8:** Coefficient of combined loss $\alpha$ and $\beta$ of each task

## A.4 DIFFTACTILE TASK AND EVALUATION DETAILS

### A.4.1 TASK SETUP DETAILS

**Reinforcement Learning (RL)**   We use object particles' state vector $o_i(t)$ and tactile sensor's pose $T_s(t)$ as state observations. Additionally, we use tactile markers' position in 2D image $m_i = (u_i, v_i)$, three-axis contact force $F(t) = (F_x, F_y, F_z)$, and contact location center $l(t)$ by averaging all in-contact nodes' locations as tactile observations. Given that the total number of markers is 136, we downsample the number of object particles to four times of the number of markers, ensuring a balanced dimensionality across different input segments. The input vector is formulated as either only state observation or with additional tactile observation. Then it is fed into a Multi-Layer Perceptron (MLP) policy network.

We use stable-baseline3 (Raffin et al., 2021)'s default PPO and SAC as our policy networks. Given an initial trajectory which is the same for all baseline methods, the policy network takes the input vector and outputs an action $\Delta v_s(t)$ of the sensor for each time step. Then we update the sensor's velocity as $v_s(t) + = \Delta v_s(t)$. We constraint the actions in the range of $[-0.15, 0.15]$ for a reasonable action size.

**CMA-ES**   In each optimization step, we generate 20 new trajectories based on the current trajectory with a standard deviation of 0.15 for a fair comparison with RL. We evaluate each new trajectory's loss and then update the policy based on the evaluation. This then informs the generation of the next optimization step's 20 trajectories. We used the same initial trajectory as RL and ran 100 optimization steps in total for each task.

**Gradient-based Optimization (Ours)**   , In each optimization step, we forward the simulation and calculate the defined loss, and then backpropagate the gradients from the loss to the target optimization variables. We then update the target variables with Adam optimizer. To enhance optimization efficiency, we use different learning rates for different optimization variables. The hyper-parameters can be found in Table 7, where $lr_p$ is the learning rate for translation, $lr_o$ is the learning rate for orientation, and $lr_w$ is the learning rate for the gripper's width. Note that for tasks where we use a single tactile sensor, the value of $lr_w$ is listed as N/A.

### A.4.2 LOSS AND REWARD

During training, we assign task-specific weights to state and tactile losses, denoted as $\alpha$ and $\beta$. The final loss is then calculated as $L_{total} = \alpha \times L_{state} + \beta \times L_{tactile}$. The task-specific values for $\alpha$ and $\beta$ are provided in Table 8. We use the losses discussed in Section 4.5 and Section 4.6 for optimization-based methods including our gradient-based method and CMA-ES; for model-free RL algorithms, we subtract the cumulative loss of two consecutive steps to obtain each step's loss, and then calculate the reward to fit the settings of RL algorithms.

### A.4.3 METRICS DETAILS

We design task-specific metrics for evaluations. Metrics' mathematical formulas are shown in Table 10.

| Notion | Explanation |
|--------|-------------|
| $P(t)$ | 3D position of the center of the object |
| $F(t)$ | Aggregated three-axis force vector on the surface of the tactile sensor |
| $F_t(t)$ | The shear force in the sensor coordinate frame |
| $F_n(t)$ | The normal force in the sensor coordinate frame |
| $\mu$ | The friction coefficient |
| $l(t)$ | The center location of the contact area on the tactile sensor surface |
| $\theta(t)$ | The rotated angle of the object from its initial pose |
| $SDF(t)$ | The signed distance field of the object |
| $M(t)$ | The mass distribution of the object |
| $d_{contact}(t)$ | The traveling distance of the sensor while being in contact with the surface |
| $N$ | The number of the object's particles |

**Table 9:** Explanation of parameters used in loss and metric computation

**Surface Follow**  We evaluate the continuous in-contact distance $d_{contact}(t)$ the sensor travels on the surface within a fixed timestep span. Here, a longer distance means better results.

**Cable Straighten**  Our metric is the average displacement of each particle $i$ on the cable from its target horizontal position $||p_i(t) - p_i(target)||$. A smaller value indicates a more desired result.

**Case Open**  We calculate the opened angle in degrees between the case lid and the horizontal tabletop $\theta(t)$. The opened angle, due to gravity, can potentially show a negative value if the training results are suboptimal. Therefore, a larger value suggests better performance.

**Object Repose**  We measure the rotated angle in degrees of the object $\theta(t)$ from its initial pose. A larger value in this context indicates better performance.

| Task | Mathematical Formula of the Metric |
|------|-----------------------------------|
| SurfaceFollow | $d_{contact}(t)$ |
| CableStraighthen | $\frac{\sum_i ||p_i(t) - p_i(target)||^2}{N}$ |
| CaseOpen | $\theta(t)$ |
| ObjectRepose | $\theta(t)$ |

**Table 10:** Metrics for four contact-rich manipulation tasks.

### A.4.4  LOSS DETAILS

We design different losses used for different tasks to obtain state or tactile reward, shown in Table 11.

**Grasping**  The losses we used are defined in Section 4.5. $\gamma$ and $\eta$ are set to 0.1.

**Surface Follow**  We use $L_{pos}$ as the state loss and $L_{force}$ as the tactile loss.

**Cable Straighten**  We use $L_{cable}$ as the state reward which is the sum of the distance between the target position and the current position for each node on the cable. Tactile loss comprises $L_{force} + L_{loc}$.

**Case Open** & **Object Repose**  We use $L_{angle}$ as the state loss and $L_{force}$ as the tactile loss.

### A.4.5  PARAMETER DETAILS

We apply optimized parameters including FEM-based sensor's Lamé parameters $\mu$ and $\lambda$ in manipulation tasks since we only use one kind of tactile sensor. Other parameters vary depending on tasks since different tasks use different objects. Sensor-related parameters are shown in Table 12. Object simulation parameters for different tasks are shown in Table 13, where $S_{obj}$ is the object scale compared to the grid size in MLS-MPM, $\rho$ is the density, $N_p$ is the number of particles in one dimension of the space, $\mu$ and $\lambda$ are Lamé parameters, and $\sigma$ is the yield stress for the elastoplastic

| Loss | Equation |
|---|---|
| $L_{pos}$ | $||P(t) - P_{target}||^2$ |
| $L_{deform}$ | $\gamma L_{dist} + \eta L_{mass}$ |
| $L_{dist}$ | $SDF(t) \cdot SDF_{target}$ |
| $L_{mass}$ | $||M(t) - M_{target}||$ |
| $L_{slip}$ | $\frac{||F_t(t)||}{\mu||F_n(t)||}$ |
| $L_{force}$ | $||F(t) - F_{target}||^2$ |
| $L_{loc}$ | $||l(t) - l_{target}||^2$ |
| $L_{angle}$ | $||\theta(t) - \theta_{target}||^2$ |
| $L_{cable}$ | $\sum_i ||p_i(t) - p_i(target)||^2$ |

**Table 11:** Losses used for manipulation tasks.

| Task | $\mu$(FEM)/$Pa$ | $\lambda$(FEM)/$Pa$ | $k_n$(Contact) | $k_d$(Contact) | $k_t$(Contact) | $\mu$(Contact) |
|---|---|---|---|---|---|---|
| **ObjectRepose** | $1.294e^3$ | $9.201e^3$ | 55.0 | 269.44 | 108.72 | 14.16 |
| **CableStraighthen** | $1.294e^3$ | $9.201e^3$ | 55.33 | 239.97 | 94.35 | 4.90 |
| **CaseOpen** | $1.294e^3$ | $9.201e^3$ | 34.53 | 269.44 | 108.72 | 14.16 |
| **SurfaceFollow** | $1.294e^3$ | $9.201e^3$ | 34.53 | 269.44 | 154.78 | 43.85 |
| **Grasping** | $1.294e^3$ | $9.201e^3$ | 55.33 | 239.97 | 94.35 | 4.90 |

**Table 12:** FEM-based sensor parameters and contact model parameters.

object. For the articulated object, we list the Lamé parameter for parts from the top to the bottom of the object.

### A.4.6 TRAJECTORY OPTIMIZATION TRAINING LOSS CURVES

We show the trajectory optimization training loss curves for four contact-rich manipulation tasks in Fig. 7. For each task, we set 100 optimization iterations for fair comparison. From the curves, we show state + tactile settings converge faster than the state-only settings which indicates the benefits of using tactile information on these manipulation tasks.

### A.4.7 DISCUSSIONS ON CONTACT-RICH MANIPULATION TASKS

For RL algorithms, both PPO and SAC's losses did not decrease in 100 iterations. There are three potential reasons: 1. The amount of data we used is insufficient for RL algorithms to learn within 100 iterations. 2. Our tasks include continuous contact, whereas RL algorithms operate on a discretized per-small-time-step basis, making it challenging to optimize. 3. RL algorithms in general require more detailed reward designs while the current reward functions are too simple to train proper policies.

For CMA-ES, we find that training losses decreased slowly. This is because CMA-ES is a sample-based optimization method and it is not as efficient as the gradient-based optimization method. It

| Task | Object Type | $S_{obj}$ | $\rho/(g/cm^3)$ | $N_p$ | $\mu$ /$Pa$ | $\lambda$/$Pa$ | $\sigma$/$Pa$ |
|---|---|---|---|---|---|---|---|
| **ObjectRepose** | Rigid | 4.0 | 1.2 | 38 | $1.428e^3$ | $5.714e^3$ | N/A |
| **CaseOpen** | Articulated | 6.0 | 1.2 | 57 | $1.428e^3/e^1/e^5$ | $5.714e^3/e^1/e^5$ | N/A |
| **SurfaceFollow** | Rigid | 8.0 | 32.0 | 76 | $1.428e^6$ | $5.714e^6$ | N/A |
| **Grasping Elastic** | Elastic | 2.0 | 1.2 | 19 | $1.428e^3$ | $5.714e^3$ | N/A |
| **Grasping Elastoplastic** | Elastoplastic | 2.0 | 1.2 | 19 | $1.428e^3$ | $5.714e^3$ | $5e^3$ |

**Table 13:** MLS-MPM based object simulation parameters

Losses-Iteration Training Curve of Our Method

**Figure 7:** Training loss curves of trajectory optimization method for four contact-rich manipulation tasks. The top row shows the training with state-only observations and the bottom row shows the training with state+tactile observations.

| Frame per second (FPS) | Forward | Backward | FEM | Contact | MPM/PBD |
|---|---|---|---|---|---|
| **ObjectRepose** | 13.11 | 8.93 | 91.38 | 199.85 | 18.38 |
| **CableStraighthen** | 21.63 | 12.10 | 35.01 | 267.41 | 419.14 |
| **CaseOpen** | 7.41 | 4.13 | 67.07 | 39.58 | 11.46 |
| **SurfaceFollow** | 25.03 | 18.75 | 30.88 | 297.08 | 592.01 |

**Table 14:** Computational runtime benchmark on four contact-rich manipulation tasks. Simulation (Forward), Gradient backpropagation (Backward), and each module's runtime during forward simulation are reported in averaged frame per second (FPS) over a trajectory.

does show the ability to optimize the trajectory but requires more than 100 iterations of optimization to converge.

Thus, we can conclude that our proposed gradient-based optimization method with differential physics has these merits: 1) Better data usage efficiency, 2) faster converge speed with the guidance of gradients, and 3) simpler loss function design. We additionally visualize the failure cases of RL algorithms and CMA-ES on our project website.

## A.5    COMPUTATIONAL RUNTIME OF THE SYSTEM

We report the averaged computational running speed of our system on four contact-rich manipulation tasks in Table. 14. From the table, we show the simulation speed (Forward) depends on different task settings and is significantly affected by the object simulation. The gradient backpropagation (Backward) speed is twice as slow as the simulation. This is because we optimize our system to be memory efficient. During the forward simulation, we save the states of each first simulation substep, and then during each backward step, we retract them and replay the corresponding forward step to fill in the rest substeps' states. FEM-based tactile sensor simulation can consistently run at high speed (greater than 30 FPS) even with two sensors in the CableStraighthen task. However, the simulation speed of objects varies such as the multi-material MPM-based object simulation is slower than others in the CaseOpen task, while the PBD-based cable simulation is the fastest. All experiments were conducted on a Ubuntu 18.04 with AMD Ryzen 7 5800x 8-core processor and Nvidia GeForce RTX 3060.

### A.6    REAL-WORLD EXPERIMENTS

#### A.6.1    EXPERIMENTAL SETUP DETAILS

We use a Gelsight tactile sensor for both system identification tasks and sim-to-real tasks. The sensor is manufactured in the laboratory with design flexibility. The soft elastomer is made with SYLGARD 184 silicone elastomer, and in a dome shape with an inner radius of 7.5 mm and an outer radius of 15.0 mm. The sensor uses an Arducam 180-degree fisheye camera to output tactile images.

#### A.6.2    EXPERIMENTAL RESULTS

To demonstrate our proposed simulator's fidelity, we conduct two sets of experiments in the real world. First, we demonstrate that the trajectories optimized with differential physics can be deployed on real-world setups in Fig. 8. Here we show the sim-to-real transfers on SurfaceFollow and CaseOpen tasks. Then we further evaluate with a closed-loop grasp task by only using tactile sensing feedback. We train a grasp stability prediction network in simulation by using a sequence of tactile observations during grasping as inputs and predicting a binary output to indicate whether it is a stable grasp or a slippage. The prediction is then used to guide the grasp adjustment. We directly use the trained model on a real-world deformable object grasp.

To train the grasp stability prediction network, we apply domain randomization and generate multiple trajectories in simulation with different parameter settings to improve the generalization of sim2real transfer. The process of trajectory generation is we let the gripper close at the speed of $v_{close} = 5$ mm/s until the gripper begins to squeeze the object for $T_{contact}$ frames, then we let the gripper lift the object for $T_{lift}$ frames at the speed of $v_{lift} = 1$ mm/s. If the slipping distance between the object and the sensor is less than 0.75 mm, we label the trajectory as a stable grasp. The parameters we used are shown in Table 15, where $S_{obj}$ and $\rho$ are the scale and density of the object. Additionally, the object shape is chosen randomly from the object set of grasping experiments in Section 4.5. For the Lamé parameters of the tactile sensor, we use the results from system identification in Section 4.3 The Lamé parameters of the object are set as $\mu = 1.428e^3$, $\lambda = 5.741e^3$. The frequency of the system is 40 Hz.

We generate 50 trajectories of stable grasp and 50 trajectories of unstable grasp in total. We split the training/validation set in a ratio of 7:3. We use two LSTM layers and one MLP layer as the network architecture. We use the sequence of tactile markers' 2D motions as inputs for the model. Each frame's tactile markers' 2D motions are obtained by subtracting the initial marker positions from the marker positions in the current frame. Due to the varying number of frames in different trajectories, we perform zero-padding at the beginning of the trajectories, making the network input size (L, 136 × 2), where L is the maximum number of frames among these trajectories. After training for 10 epochs, the success rate reaches 94.3% on the training set and 90.0% on the validation set.

We present our sim2real adaptive grasp policy for grasping a deformable object. Our goal is to grasp the object with minimal deformation. We applied our trained model directly on the real-world setup to perform an adaptive grasp. We first attempt to grasp and lift the object with minimal force, feeding the sequence of tactile marker motions into our trained model. If the model predicts slippage, we tighten the gripper, or if the model predicts a stable grasp, we continue lifting the object to the determined height. In Fig. 9, we show a comparison of our approach with two baselines: 1) forceful grasp where the gripper tightly grips the object, and 2) light grasp where the gripper lifts the object upon contact. We can see that our method successfully grasps and lifts the deformable object with minimal deformation while the two baselines failed by damaging the object or causing slippage.

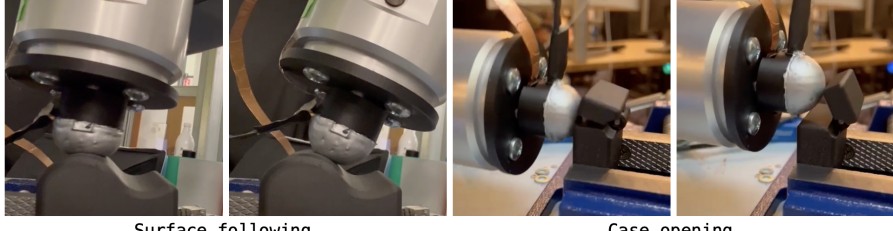

**Figure 8:** Real-world experiments of surface following and case opening tasks. We deploy the optimized trajectories from simulation directly to real-world setups.

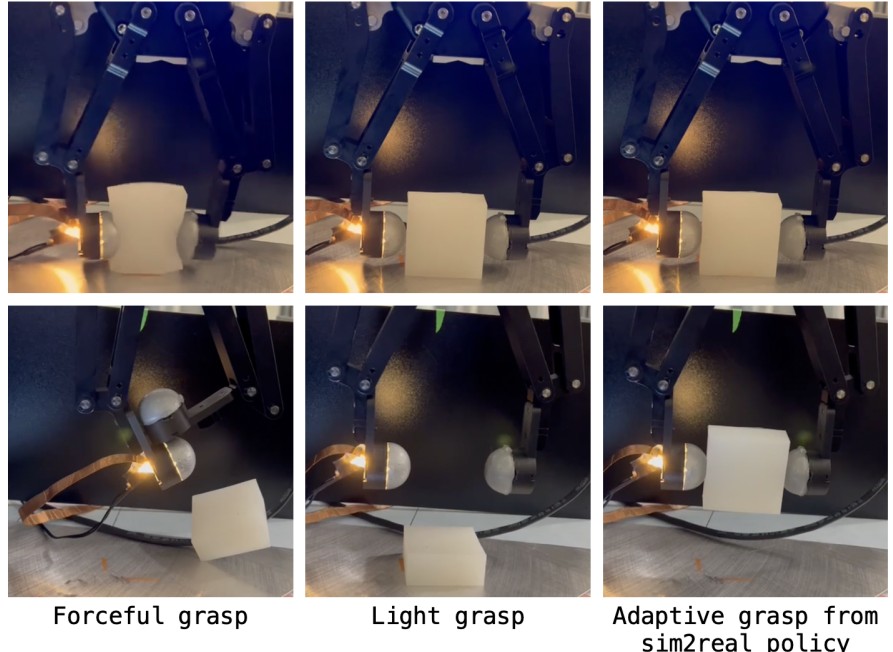

**Figure 9:** Real-world experiments of grasping a deformable object. With hardcoded policies such as forceful grasp or light grasp, they failed to grasp the deformable object with either damage or slippage. With our sim2real adaptive grasp policy, we can successfully grasp the object.

| Parameter | Value |
|:---:|:---:|
| $k_n$ | [10,100] |
| $k_d$ | [100,400] |
| $k_t$ | [100, 250] |
| $\mu$ | [5, 80] |
| $S_{obj}$ | [2, 5] |
| $\rho$ | [0.1, 4] |
| $T_{contact}$ | [0, 60] |
| $T_{lift}$ | 60 |

**Table 15:** Parameters for domain randomization on sim2real grasp policy training.

