# OpenReview forum: "DIFFTACTILE: A Physics-based Differentiable Tactile Simulator for Contact-rich Robotic Manipulation"
_ICLR.cc/2024/Conference — ICLR 2024 poster_

### Official Review · Reviewer_Zvck · 2023-10-30

**Soundness:** 3 good
**Presentation:** 4 excellent
**Contribution:** 3 good
**Rating:** 8
**Confidence:** 3

**Summary:**

The paper presents a new tactile simulation framework for soft optical sensors applied on robotics manipulation tasks. This fully-differentiable pipeline is then used for system identification of sensor properties, and gradient-based trajectory optimization. This framework is the first differentiable simulation that supports both soft optical sensors and soft object interaction.

**Strengths:**

The authors positioned the framework well in context to the state of the art, and it is clear what improvements the new simulation offers. The experiments show why the differentiability is beneficial and how the optical simulation compares to previous work and on real-world experiments. A wide range of tasks were tested to verify the results for the optical sensors, and show a significant advantage of using differentiable tactile simulations.

**Weaknesses:**

1. While the accuracy of the optical sensor model is verified, the simulation itself would benefit from having a comparison to real-world experiments as well. Questions such as how well the dynamic behavior of soft objects match reality (bouncing objects), or how well the contact model applies to objects sliding/being pushed under friction. These would be more of a benchmark on the FEM-MPM-PBD simulation, but this could nevertheless broaden the applicability of the framework.
2. How does the simulation compare when simulating articulated bodies? For example the lid-opening task, is the hinge a soft body or a joint? How well does this compare to modeling real-world objects?
3. Adding some runtime reports, at least in the appendix, would be appreciated from a practitioner's perspective, since the choice for using FEM only for the sensor likely stems from high accuracy but high computational complexity as well. Hence MPM or PBD was used to simplify object simulation, is this correct? If so, it would be interesting to see how expensive each part of the simulation is, where simplifications are necessary when used in practice. Were any of the learned grasping or manipulation policies applied on the real robot?
4. Continuing on the topic of runtimes, for the trajectory optimization tasks in manipulation, it would be interesting to see how many iterations/computational resources each method was given to converge, was it the same for each, or was each method run until convergence?

**Questions:**

1. Were the constraints applied to the reinforcement learning methods for trajectory planning also applied to the gradient-based optimization?
2. In Table 1, should the Tacchi method not also be differentiable since they are implemented in Taichi?
3. How efficient is it to simulate rigid objects as elastic using MPM and then applying rigidity constraints? Are there any plans on extending the simulator to use rigid-body or articulated-body solvers?
4. Are the results found from the SysID parameters used for the follow-up tasks?

---

> ### Author Response · Authors · 2023-11-19
> **Response to Reviewer Zvck - Part 1**
>
> We thank reviewer Zvck for acknowledging our contributions and here is our response regarding the reviewer’s questions and concerns:
>
> 1. The simulation itself would benefit from having a comparison to real-world experiments as well. Questions such as how well the dynamic behavior of soft objects matches reality (bouncing objects), or how well the contact model applies to objects sliding/being pushed under friction.
>
> Thank you for the thoughtful feedback. We have added real-world experiments to further evaluate our system in Section A.6.2. We also added the real2sim system identification demonstrations on our website to show the comparison between sim and real for pressing, sliding, and twisting scenarios. We believe simulating objects is essential for manipulation, several works such as Pac-Nerf [https://arxiv.org/abs/2303.05512], NCLaws [https://arxiv.org/abs/2304.14369] have presented the accuracy of soft object simulation with sim-to-real comparison. However, the focus of our work is the tactile sensor simulation and since our simulation system is modulized, it is straightforward to replace one or multiple modules with improved simulation approaches such as simulating objects with FEM or using constraint-based contact models. We would like to explore the fidelity of the different object simulation methods in future work.
>
> 2. How does the simulation compare when simulating articulated bodies? For example the lid-opening task, is the hinge a soft body or a joint? How well does this compare to modeling real-world objects?
>
> We approximate the articulated objects by using the MPM-based approach and assign different materials for different parts. Here the joint is simulated as a soft and thin body, and other parts are simulated as rigid bodies.  We further demonstrated that the optimized trajectory for the case opening task can be successfully deployed on real-world settings with a real articulated object in Section A.6.2.
>
> 3. Adding some runtime reports, at least in the appendix, would be appreciated from a practitioner's perspective. MPM or PBD was used to simplify object simulation, is this correct? If so, it would be interesting to see how expensive each part of the simulation is, where simplifications are necessary when used in practice. Were any of the learned grasping or manipulation policies applied on the real robot?
>
> We have added the running time report in Section A.5 on all four contact-rich tasks. We provide both the simulation time and gradience backpropagation time, along with each simulation module (FEM, contact, MPM/PBD) running time in Table 11.
>
> And yes, MPM and PBD are used to simplify object simulation, we do observe fast PBD simulation speed but MPM simulation speed is rather slower because of the large amount of particles we used in simulation.
>
> We also added two sets of real-world experiments to further evaluate our system in Section A.6.2. First we deploy the optimized trajectory from simulation on real-world robot setup for surface following and box opening tasks. Then we train a grasp stability prediction model in simulation based on tactile sensing data.  We directly apply the trained model to a real-world deformable object grasp task and show that we can adaptively grasp the soft deformable object.
>
> 4. For the trajectory optimization tasks in manipulation, it would be interesting to see how many iterations/computational resources each method was given to converge, was it the same for each, or was each method run until convergence?
>
> We have added the training curves of trajectory optimization on contact-rich manipulation tasks in Section A.4.6. To make a fair comparison with other baselines, we run all experiments with 100 iterations of optimization. From the training loss curves, we see most tasks can converge within 100 iterations.
>
> 5. Were the constraints applied to the reinforcement learning methods for trajectory planning also applied to the gradient-based optimization?
>
> Yes. Since our system is implemented with semi-explicit solvers which require a small simulation step size, we constrain the action space to [-0.15, 0.15] for RL and set small learning rates for gradient-based methods to ensure the stability of the simulation. And we use the same initial trajectories for both the gradient-based method and RL methods.

---

> ### Author Response · Authors · 2023-11-19
> **Response to Reviewer Zvck - Part 2**
>
> 6. In Table 1, should the Tacchi method not also be differentiable since they are implemented in Taichi?
>
> It is true that Tacchi is implemented with Taichi, and it can be differentiable in theory. However, the authors of Tacchi didn’t implement the gradient backpropagation which is non-trivial and didn’t show any trajectory optimization with differential physics.
>
> 7. How efficient is it to simulate rigid objects as elastic using MPM and then applying rigidity constraints? Are there any plans on extending the simulator to use rigid-body or articulated-body solvers?
>
> We admit that it is not the most efficient way to simulate rigid objects by using the MPM method as we report the runtime of object simulation in Table 11. For example, we use a rigid object in the object reposing task, and due to dense particles, the object simulation is the runtime bottleneck. We would like to further extend our simulation system to support rigid-body and articulated-body models.
>
> 8. Are the results found from the SysID parameters used for the follow-up tasks?
>
> Yes, we use the identified tactile sensor parameters from Real2Sim for all following manipulation tasks. However, contact parameters such as the surface friction coefficient also depend on object materials. But these can still serve as good references and we use them by adding randomization based on the identified parameters. For object parameters, we randomize them within a range based on the tactile sensor's and contact model’s parameters to make sure the system can stably run.  All parameter details are added in the Section. A.4.5.
>
> We hope these addressed the concerns and please let us know if there are any further questions!
>
> Thanks,
>
> Authors

---

> ### Comment · Reviewer_Zvck · 2023-11-21
>
> Thank you for the extensive replies to all the reviewers, and for grouping the concerns into specific categories. The extra runtime information in the appendix helps with the clarity, and it is now also clear what the limitations and future work are in this domain. The additional real robot experiments make the paper much stronger, and the simplifications made in simulation on some objects, like the articulated robot or rigid robots in general, seems to not impact the sim-to-real transfer too much. Very impressive work, thanks again to the authors.

---

> > ### Author Response · Authors · 2023-11-22
> > **Response to reviewer Zvck**
> >
> > We really appreciate your recognition of our contributions and thank you again for your time and effort in reviewing our work!

---

### Official Review · Reviewer_EisY · 2023-10-31

**Soundness:** 3 good
**Presentation:** 2 fair
**Contribution:** 3 good
**Rating:** 6
**Confidence:** 3

**Summary:**

The paper presents DiffTactile, a simulator that uses FEM to simulate soft tactile sensors such as Gelsight, supports elastic, rigid, elasto-plastic, and cable-like objects, and is differentiable. The authors demonstrate the use of the simulator for three tasks. First, in system identification, the goal is to use real-world tactile observations to optimize the simulator’s physical parameters, and then compare the tactile marker and force reading to that of real data. In optical simulation, the visual quality of the optical simulation is examined. In grasping, the task is to use a parallel jaw gripper equipped with tactile sensors to see if tactile feedback is helpful. Finally in manipulation, four tasks are performed: surface following, cable straightening, case opening, and object reposing.

**Strengths:**

- Developing accurate simulation for tactile sensors is quite challenging, particularly for optical tactile sensors. However, the demand is quite large, as it can be expensive or challenging to work with real tactile sensing hardware. This simulator has many features including simulation of many object types and optical simulation, which are not present in existing works and may make it valuable for the development of robotic tactile sensor applications.
- The optical simulation results look quite impressive and match well with the real readings.
- The introduction of the manipulation tasks is a nice demo of the types of tasks that can be modeled and learned in this simulator.

**Weaknesses:**

- In the system identification task, it seems like the pixel-wise tactile marker mean squared error is extremely high for the real to sim setting, or at least, the standard deviation or standard error (which one is it? It’s not labeled) is much larger than the differences between the different methods, including the random method. This is rather concerning, as it seems to indicate that the system identification method is not very effective at reducing the sim-to-real gap.
- Rather than a comparison between using or not using tactile sensing for grasping, which has been validated in prior works, or comparison between different methods for optimizing policies to solve the manipulation tasks, I wish the paper focused more on evaluating how realistic and accurate the tactile sensing simulation is, as well as sim to real applications. I think those are the things that will really impact whether or not practitioners can rely on this simulator to generate conclusions that will hold in the real world.
- The organization of the paper is slightly confusing: I think it would be easier to understand if the tasks were introduced closer in the text to where the results are presented. In general I think the clarity of the writing could be improved, for example, to be more explicit about when tactile signals are real or simulated (for example, in Section 4.2, is the training data real?)

**Questions:**

Please see my points in "weaknesses". In addition:
- What differentiates the “grasping” task from the “manipulation” tasks? I think that the manipulation tasks are nice demonstrations of the types of tasks that can be simulated using DiffTactile, but I don’t quite understand what the “grasping” task is trying to show that is not being illustrated by the “manipulation” tasks, as it seems to me that it could easily become the fifth “manipulation” task.
- Can you provide some intuition or qualitative visualization for why the baselines like PPO and SAC don’t perform well on the manipulation tasks? What are the failure modes?
- (nit): I recommend adding periods after paragraph section headings or otherwise distinguishing them from the following text.
- (nit): For the “experimental results” on page 9: “For case opening and object reposing, we define the metric as the opened angle of the lid and the orientation of the object.” I think it’s not quite accurate to refer to the orientation of the object as a metric.

---

> ### Author Response · Authors · 2023-11-19
> **Response to Reviewer EisY**
>
> We thank reviewer EisY for acknowledging our contributions and here is our response regarding the reviewer’s questions and concerns:
>
> 1. In the system identification task, it seems like the pixel-wise tactile marker mean squared error is extremely high for the real to sim setting, or at least, the standard deviation or standard error (which one is it? It’s not labeled) is much larger than the differences between the different methods, including the random method.
>
> We have labeled them as standard deviations. We admit the real-2-sim errors are higher than sim-2-sim which is normally because of the real-world sensor noise. Note that 1 pixel in the tactile image space roughly equals 0.113 mm in physical space, thus our real-2-sim is still at the submillimeter level. The mean squared errors are averaged over all markers from all frames, and we see the consistency of our lower errors compared to the baselines. We also observe the same scale consistency of standard deviations. Although standard deviations are higher than the differences between the different methods, these are attributed to the real-data noise level since our sensor is manually manufactured in the laboratory.
>
> We also added real-world experiments and demonstrated the trajectories optimized in simulation by using the real-2-sim results can be directly deployed on real-world setups. To further improve the real-2-sim results, collecting more data for system identification or using commercialized sensors can be explored as future work.
>
> 2. I wish the paper focused more on evaluating how realistic and accurate the tactile sensing simulation is, as well as sim to real applications.
>
> We have added two sets of real-world experiments to further evaluate our system in Section A.6.2. First we deploy the optimized trajectory from simulation on real-world robot setup for surface following and box opening tasks. Then we train a grasp stability prediction model in simulation based on tactile sensing data.  We directly apply the trained model to a real-world deformable object grasp task and show that we can adaptively grasp the soft deformable object.
>
> 3. I think it would be easier to understand if the tasks were introduced closer in the text to where the results are presented. In general I think the clarity of the writing could be improved, for example, to be more explicit about when tactile signals are real or simulated
>
> Thank you for the writing suggestions, we have updated the manuscript accordingly. We re-arranged the task description at the beginning of Section 4 Experiments and added clarifications for the experiments.
>
> 4. What differentiates the “grasping” task from the “manipulation” tasks?
>
> We have merged the “grasping” as one of the manipulation tasks. However since we have variations on the grasping task, we kept it stand-alone and renamed the rest four tasks as “contact-rich manipulation tasks”.
>
> 5. Can you provide some intuition or qualitative visualization for why the baselines like PPO and SAC don’t perform well on the manipulation tasks? What are the failure modes?
>
> We have added the discussions of the experiments in Section A.4.7. Our system is implemented with semi-explicit solvers which require a small simulation step size as a common strategy. Therefore, in our RL experiments, we did not allow the policy to search randomly but within a constraint action space. We initialized the trajectories as the same as the ones used for trajectory optimization and used the same reward functions. This ensures the fairness of comparison. However, after training for 100 iterations, both PPO and SAC’s training losses did not decrease.
> We intuitively attribute this to:
> a. The amount of data we used is insufficient for RL algorithms to learn within 100 iterations.
> b. Our tasks include continuous contact, whereas RL algorithms operate on a per-small-time-step basis, making it challenging to optimize.
> c. RL algorithms in general require more detailed reward designs while the current reward functions are too simple to train proper policies.
>
> We also added visual demonstrations of the failure cases of RL approaches on our website: https://difftactile.github.io/.
>
> 6. I recommend adding periods after paragraph section headings or otherwise distinguishing them from the following text.
>
> Thanks for the writing suggestions and we have added space between section headings and texts.
>
> 7. For the “experimental results” on page 9: “For case opening and object reposing, we define the metric as the opened angle of the lid and the orientation of the object.” I think it’s not quite accurate to refer to the orientation of the object as a metric.
>
> Thank you for the writing suggestions, we have rephrased and clarified the sentence.
>
> We hope these addressed the concerns and please let us know if there are any further questions!
>
> Thanks,
>
> Authors

---

> > ### Author Response · Authors · 2023-11-22
> > **Response to Reviewer EisY**
> >
> > Hi reviewer EisY, thanks again for your time and effort in reviewing our work! We would like to kindly remind you that the discussion phase is reaching its end, and we would like to hear any feedback from you!

---

> > ### Comment · Reviewer_EisY · 2023-11-22
> > **Response to authors**
> >
> > Thank you for your efforts in providing additional experiments and adjusting the writing. I particularly appreciate the addition of the real-world experiments, however, I agree with reviewer TYvh that it is hard to tell whether the performance of the trajectory in the video is due to successful system identification and that additional quantitative and qualitative comparisons to other system ID methods would support the hypothesis much more strongly here.
> >
> > Thank you for the clarification regarding the performance of the RL experiments. It is understandable that having a small time step would make it challenging for those methods to perform well. However, I don't quite understand why it is necessary to not allow the policy to search randomly but instead within a constrained action space. Could you please elaborate on that? It would be helpful if this were clarified in the paper to understand the particular setting of the RL experiments for future practitioners. I could imagine that a carefully tuned RL configuration could learn to solve these tasks as well, and while I understand that differentiability is a key feature of DIFFTACTILE, it seems that the additional learnability of these tasks with RL would only be a plus. The contributions on the simulation side are significant and strong in my opinion, but seems orthogonal to the message that RL simply works more poorly than gradient-based optimization (which is what Table 5 seems to be showing).
> >
> > I will keep my score for now, but am willing to increase it given further evidence from real-world experiments.

---

> ### Author Response · Authors · 2023-11-23
> **Response to Reviewer EisY**
>
> Hi reviewer EisY,
>
> Thanks again for your insightful feedback to strengthen our work!
>
> Regarding the RL action space searching:
> - We use a penalty-based contact model to couple the FEM-base sensor with MPM/PBD-base objects. That means during each simulation step, we first update both the mesh nodes of the sensor and the particles of the object, if penetration happens, we push the particles out of the mesh surface to bound the contact. If the action step is too large, the penetrations would be too aggressive which will lead to the flipped FEM element volumes and the system will crash. To keep the stability of the system, we need to constraint to small actions. We admit this is one limitation of the semi-explicit system. It is possible to resolve this issue by switching to an implicit system that is more tolerant of the large simulation step size. However, we would like to highlight the advantages of the semi-explicit system: the implementation simplicity especially for the gradient backward, faster running speed for small-scale systems (such as thousands of mesh nodes for FEM as our current settings), and easier to customize and adapt to various tasks.
>
> Regarding the real-world experiments:
> - We have incorporated a new comparison with a baseline system identification method on the real-world box opening task as visualized on our website. Notably, the baseline method predicted a higher Young's modulus value, indicating a stiffer sensor material. This leads to the optimized trajectory exhibiting lighter force application during the box opening, therefore the policy fails to transfer to the real world. We believe that our real-world experiments have shown the potential of our simulators.
> - While we demonstrated the role of system identification in enhancing sim-to-real transfer, there exist alternative approaches to bridge sim-to-real gaps, such as data augmentation and domain randomization. Our proposed simulator provides a platform to explore these approaches, as one part of our contribution. System identification, in our perspective, acts as a "helper" rather than a "unique enabler" in minimizing these gaps. Thus, while we appreciate the value of using sim-to-real downstream tasks to assess system identification, the real2sim2real is only one application of our simulation and the fundamental question we are trying to answer here is whether the policy optimized in our simulator can be combined with existing technique to enable sim2real transfer.
> - We would like to emphasize that our main contributions of this work are providing a tactile simulator that supports differential physics, optical simulation, a broad range of object manipulation, and demonstrating various manipulation tasks in both simulation and real world. System identification is only a part of our contribution. To the best of our knowledge, our tactile simulator is the first to integrate all these features. We wish the reviewer to reevaluate our contribution based on the broader significance of our work.
>
> Best regards,
>
> Authors

---

> > ### Comment · Reviewer_EisY · 2023-12-01
> > **Response to authors**
> >
> > I thank the authors for their detailed responses and efforts in providing the requested experiments.
> >
> > I understand that computational restraints make it so that the simulation step size must be smaller, however, it seems that it would be reasonable in this situation to temporally abstract the environment for the sake of running reinforcement learning (e.g. by repeating the same "action" for multiple simulation steps). Nevertheless, as I mentioned before, I feel that the additional learnability of these tasks with RL would only be a plus and that the message that RL simply works more poorly than gradient-based optimization feels orthogonal to the main contribution of the paper (a very impressive simulator). I hope that in future versions this can be integrated into the writing.
> >
> > I also appreciate the addition of a qualitative baseline for the real world system ID experiments. Although a quantitative evaluation would be more convincing, I understand that time constraints may have made it difficult to complete them.
> >
> > Overall, I have increased my score to 6 to vote for weak acceptance.

---

### Official Review · Reviewer_Wssn · 2023-11-01

**Soundness:** 3 good
**Presentation:** 3 good
**Contribution:** 3 good
**Rating:** 6
**Confidence:** 1

**Summary:**

This paper is about creating a differentiable tactile stimulator that supports contact rich tasks. Having such a simulator is important to learn robust policies using tactile. The simulator consists of 4 components: 1) sensor deformation through FEM 2) Optical simulation model that maps sensor deformation to rgb reflected color. 3) Objects are modeled using least square material point and position based dynamics. 4) penalty based contact model which goes from deformation, represented by SDF, to forces. They evaluated the simulator in 3 tasks: system identification which basically tries to estimate the sensor params from a set of collected real sensor data. The second task is grasping fragile objects and finally manipulating non-rigid objects such as straightening a cable.

**Strengths:**

- Having differentiable simulator for contact-rich tasks unlock a lot of new capabilities.
- The paper covers experiments in wide range of applications.

**Weaknesses:**

I would have liked to see application of the method for a contact-rich manipulation task in the real world.

**Questions:**

N/A

---

> ### Author Response · Authors · 2023-11-19
> **Response to Reviewer Wssn**
>
> We thank reviewer Wssn for investing time in reviewing our work and acknowledging our contributions.
>
> As the reviewer suggested, we have added two sets of real-world experiments to further evaluate our system in Section A.6.2. First we deploy the optimized trajectory from simulation on real-world robot setup for surface following and box opening tasks. Then we train a grasp stability prediction model in simulation based on tactile sensing data.  We directly apply the trained model to a real-world deformable object grasp task and show that we can adaptively grasp the soft deformable object.
>
> We hope these addressed the concerns and please let us know if there are any further questions!
>
> Thanks,
>
> Authors

---

> > ### Author Response · Authors · 2023-11-22
> > **Responce to Reviewer Wssn**
> >
> > Hi reviewer Wssn, thanks again for your time and effort in reviewing our work! We would like to kindly remind you that the discussion phase is reaching its end, and we would like to hear any feedback from you!

---

### Official Review · Reviewer_TYvh · 2023-11-03

**Soundness:** 3 good
**Presentation:** 3 good
**Contribution:** 2 fair
**Rating:** 6
**Confidence:** 3

**Summary:**

This paper presents a differential simulator for tactile sensors which work on the principle of light reflection from coloured deformable surfaces. The deformable surface is simulated by a finite element model, with contact forces based on penetration penalty. Surface normals of the deformed tactile surface are used by a neural network to predict the RGB colour of the reflected light. All these operations are differentiable

The paper presents experiments about system identification, grasping, and various manipulation tasks - all in simulation. The differential nature of the simulator allows gradient-based trajectory optimization for these tasks. Experiments show that this outperforms CMA-ES and RL.

**Strengths:**

- Tactile sensors provide highly useful sensing streams for fine manipulation tasks. However, they have been difficult to simulate. This hampers the training of policies with tactile observations in simulation. The differentiability of the proposed simulation system can enable data-efficient system identification. Gradient-free methods can also be used for system identification, but they usually require more data, including real robot data synchronized to sim, which can be expensive.
- Multiple experiments with multiple tasks show the general applicability of the proposed system.
- The paper is well written and easy to understand.

**Weaknesses:**

- It is difficult to judge the accuracy of system identification based on the MSE in tactile markers location alone. Small errors can lead to large drops in downstream task performance. Therefore, system identification algorithms are usually evaluated by sim2real task performance [1, 2]. This paper lacks sim2real task performance experiments.
- Is CMA-ES not applicable to the system identification task (Section 4.1)? If it is, please discuss why it was not used as a baseline.
- A lack of discussion of computation time, especially the FEM-based deformation module.
- A lack of implementation details like RNN structure for parameter identification, optical prediction network architecture (Section 4.2), mathematical formulations of the reward functions used for the manipulation tasks (Section 4.4).

### References
1. "Generation of GelSight Tactile Images for Sim2Real Learning" RA-L
2. "Efficient tactile simulation with differentiability for robotic manipulation" - PMLR

**Questions:**

- What is the computational runtime of the proposed method, and how does it affect the intended applications?
- Which simulation parameters are used for the manipulation experiments in Section 4.4? Are they the parameters identified from the real robot system?

### After rebuttal
I would like to thank the authors for addressing my concerns from the review during the rebuttal phase. I am raising my rating because of this.

---

> ### Author Response · Authors · 2023-11-19
> **Response to Reviewer TYvh**
>
> We thank reviewer TYvh for acknowledging our contributions and here is our response regarding the reviewer’s questions and concerns:
>
> 1. This paper lacks sim2real task performance experiments.
>
> We have added two sets of real-world experiments to further evaluate our system in Section A.6.2. First we deploy the optimized trajectory from simulation on real-world robot setup for surface following and box opening tasks. Then we train a grasp stability prediction model in simulation based on tactile sensing data.  We directly apply the trained model to a real-world deformable object grasp task and show that we can adaptively grasp the soft deformable object.
>
> 2. Is CMA-ES not applicable to the system identification task (Section 4.1)? If it is, please discuss why it was not used as a baseline.
>
> CMA-ES is applicable to the system identification task and we have updated Table 2 with CMA-ES results as baselines. From the results, we observed that CMA-ES performs second best after our proposed method in sim2sim scenarios, outperforming other baselines. However, its performance on real2sim is worse than RNN.
>
> 3. A lack of discussion of computation time, especially the FEM-based deformation module. What is the computational runtime of the proposed method, and how does it affect the intended applications?
>
> We have added the running time report in Section A.5 on all four contact-rich tasks. We provide both the simulation time and gradience backpropagation time, along with each simulation module (FEM, contact, MPM/PBD) running time in Table 11.
>
> From the runtime reports, we show that the FEM module in the system can run greater than 30 fps for forward simulation, but the whole system’s speed also depends on object simulation and will double for the gradient backpropagation (backward). For example, the slowest task case opening requires 6 hours to optimize one trajectory with 600 simulation steps.
>
> 4. A lack of implementation details like RNN structure for parameter identification, optical prediction network architecture (Section 4.2), mathematical formulations of the reward functions used for the manipulation tasks (Section 4.4).
>
> We have added the RNN structure details in Section A.2. We also added the details of the reward functions in Section A.4.3 and A.4.4. Optical prediction network architecture will be added in Section A.3.
>
> 5. Which simulation parameters are used for the manipulation experiments in Section 4.4? Are they the parameters identified from the real robot system?
>
> We have added the simulation parameter details in Section A.4.5. Yes, we use the identified tactile sensor parameters from Real2Sim for all the following manipulation tasks. However, contact parameters such as the surface friction coefficient also depend on object materials. But these can still serve as good references and we use them by adding randomization based on the identified parameters. For object parameters, we randomize them within a range based on the tactile sensor's and contact model’s parameters to make sure the system can stably run.
>
> We hope these addressed the concerns and please let us know if there are any further questions!
>
> Thanks,
>
> Authors

---

> > ### Comment · Reviewer_TYvh · 2023-11-20
> > **reply**
> >
> > Thank you authors, for the responses and new data.
> > - **New real robot experiments**. Can you please clarify about the parameters of the simulator used to train the surface following policy and the grasp stability prediction network? Were those simulator parameters system-identified using data collected from the real robot system? In other words, do these new experiments show sim-to-real, or real-to-sim-to-real?
> > - **How to validate the task correctness of system identification?** The surface following and box opening videos on the website are quite short. Neither this nor the grasping experiment sufficiently convince the reader that the real-to-sim system identification is the unique enabler of this behaviour. What is the performance with other system identification methods like RNN and CMA-ES, which are evaluated in this manuscript?

---

> ### Author Response · Authors · 2023-11-22
> **Response to Reviewer TYvh**
>
> Hi reviewer TYvh,
>
> Thanks for your prompt reply and feedback!
>
> - The parameters we used for both new tasks are listed in our Appendix Section A4.5 and A6.2. For the surface following and box opening task, we used the system-identified parameters (in Tables 12 & 13). For the grasp stability prediction task, we added domain randomization on parameters (in Table 15) to improve the performance.  We believe the surface following & box opening tasks show the real-to-sim-to-real and the grasping task shows the sim-to-real.
> - Thanks for bringing up this great point and we do agree that task-based evaluation would be another good metric. We have been working on real-world experiment evaluation for other baselines. Based on current observations, we do see accurate system identification indeed affects real-world contact-rich robotic manipulations. For example, in the box opening task, the optimized trajectories vary with sensors with different softness and affect the success rate of policy transfer. We will upload the video results soon.
>
> Please let us know if you have any further questions!
>
> Best regards,
>
> Authors

---

> ### Author Response · Authors · 2023-11-23
> **Response to Reviewer TYvh**
>
> Hi reviewer TYvh,
>
> Thanks again for your insightful feedback to strengthen our work!
>
>
> - We have incorporated a new comparison with a baseline system identification method on the real-world box opening task as visualized on our website. Notably, the baseline method predicted a higher Young's modulus value, indicating a stiffer sensor material. This leads to the optimized trajectory exhibiting lighter force application during the box opening, therefore the policy fails to transfer to the real world. We believe that our real-world experiments have shown the potential of our simulators.
> - While we demonstrated the role of system identification in enhancing sim-to-real transfer, there exist alternative approaches to bridge sim-to-real gaps, such as data augmentation and domain randomization. Our proposed simulator provides a platform to explore these approaches, as one part of our contribution. System identification, in our perspective, acts as a "helper" rather than a "unique enabler" in minimizing these gaps. Thus, while we appreciate the value of using sim-to-real downstream tasks to assess system identification, the real2sim2real is only one application of our simulation and the fundamental question we are trying to answer here is whether the policy optimized in our simulator can be combined with existing technique to enable sim2real transfer.
> - We would like to emphasize that our main contributions of this work are providing a tactile simulator that supports differential physics, optical simulation, a broad range of object manipulation, and demonstrating various manipulation tasks in both simulation and real world. System identification is only a part of our contribution. To the best of our knowledge, our tactile simulator is the first to integrate all these features. We wish the reviewer to reevaluate our contribution based on the broader significance of our work.
>
> Best regards,
>
> Authors

---

### Author Response · Authors · 2023-11-19
**General response**

We would like to express our sincere appreciation for all the reviewers’ time and effort in reviewing our work. We are grateful for the reviewers’ recognition of our contributions to this work including
1. The unique features of our tactile simulator such as differentiability, multiple object simulation, and optical simulation, and their benefits.
2. Evaluation of the proposed system on various types of manipulation tasks.

We also thank all reviewers’ constructive and thoughtful suggestions and feedback and have revised our manuscript based on them. All changes are highlighted in red in the manuscript.

Here we list general concerns from reviewers and our responses and revisions. For the remaining concerns from each reviewer, we also provide individual responses separately below.

1. This paper lacks sim2real experiments to evaluate the accuracy/fidelity of the system.

We have added two sets of real-world experiments to further evaluate our system in Section A.6.2. First we deploy the optimized trajectory from simulation on real-world robot setup for surface following and box opening tasks. Then we train a grasp stability prediction model in simulation based on tactile sensing data.  We directly apply the trained model to a real-world deformable object grasp task and show that we can adaptively grasp the soft deformable object. We have also added visual demonstrations of the real-world experiments on our website: https://difftactile.github.io/.

2. This paper lacks the computation time reports, especially for the FEM module.

We have added the running time report in Section A.5 on all four contact-rich tasks. We provide both the simulation time and gradience backpropagation time, along with each simulation module (FEM, contact, MPM/PBD) running time in Table 11.

3. Are the parameters identified from real-world systems used in the follow-up manipulation tasks?

Yes, we use the identified tactile sensor parameters from Real2Sim for all following manipulation tasks. However, contact parameters such as the surface friction coefficient also depend on object materials. But these can still serve as good references and we use them by adding randomization based on the identified parameters. For object parameters, we randomize them within a range based on the tactile sensor's and contact model’s parameters to make sure the system can stably run. All parameter details are added in the Section. A.4.5.

Thank you,

Authors

---

### Comment · Area_Chair_zgiT · 2023-11-22

Dear all,

The author-reviewer discussion period is about to end.

@authors: If not done already, please respond to the comments or questions reviewers may further have. Remain short and to the point.

@reviewers: Please read the author's responses and ask any further questions you may have. To facilitate the decision by the end of the process, please also acknowledge that you have read the responses and indicate whether you want to update your evaluation.

You can update your evaluation positively (if you are satisfied with the responses) or negatively (if you are not satisfied with the responses or share other reviewers' concerns). Please note that major changes are a reason for rejection.

You can also keep your evaluation unchanged. In this case, please indicate that you have read the responses, that you do not have any further comments and that you keep your evaluation unchanged.

Best regards,
The AC

---

> ### Author Response · Authors · 2023-11-22
> **Response to AC**
>
> Hi Area Chair,
>
> Thank you for the kind reminder! We have replied to all reviewers and will keep being active during the rest of the discussion phase. We will try our best to address more questions from reviewers. Thank you again for your time!
>
> Best regards,
>
> Authors

---

### Meta-Review · Area_Chair_zgiT · 2023-12-09

**Metareview:**

The reviewers unanimously recommend acceptance (8-6-6-6). The paper proposes a differentiable simulator for tactile simulations. The reviewers note the general utility of the contributed simulator, underline the quality of the empirical evaluation, and appreciate the clarity of the presentation. The author-reviewer discussion has been constructive and has led to a number of improvements to the paper. We recommend the authors to address the few remaining concerns, if any, in the final version of the paper.

**Justification For Why Not Higher Score:**

The scope is at the frontier between machine learning and robotics, which may limit the interest of the paper for the machine learning community.

**Justification For Why Not Lower Score:**

The reviewers unanimously recommend acceptance.

---

### Decision · Program_Chairs · 2024-01-16

Accept (poster)